# An AuNPs-Based Fluorescent Sensor with Truncated Aptamer for Detection of Sulfaquinoxaline in Water

**DOI:** 10.3390/bios12070513

**Published:** 2022-07-11

**Authors:** Xingyue Chen, Lulan Yang, Jiaming Tang, Xu Wen, Xiaoling Zheng, Lingling Chen, Jiaqi Li, Yong Xie, Tao Le

**Affiliations:** 1College of Life Science, Chongqing Normal University, Chongqing 401331, China; chenxingyue31@163.com (X.C.); yang16086@126.com (L.Y.); tangjiamingt@163.com (J.T.); wenxu7968@163.com (X.W.); 2021110513064@stu.cqnu.edu.cn (X.Z.); cll2629593857@163.com (L.C.); lilijiaqi2020@163.com (J.L.); 2Bioassay 3D Reconstruction Laboratory, Chongqing College of Electronic Engineering, Chongqing 401331, China

**Keywords:** molecular docking, molecular dynamics simulations, aptamer truncation, fluorescence aptasensor, gold nanoparticles

## Abstract

Herein, we developed a novel truncation technique for aptamer sequences to fabricate highly sensitive aptasensors based on molecular docking and molecular dynamics simulations. The binding mechanism and energy composition of the aptamer/sulfaquinoxaline (SQX) complexes were investigated. We successfully obtained a new SQX-specific aptamer (SBA28-1: CCCTAGGGG) with high affinity (K_d_ = 27.36 nM) and high specificity determined using graphene oxide. This aptamer has a unique stem-loop structure that can bind to SQX. Then, we fabricated a fluorescence aptasensor based on SBA28-1, gold nanoparticles (AuNPs), and rhodamine B (RhoB) that presented a good linear range of 1.25–160 ng/mL and a limit of detection of 1.04 ng/mL. When used to analyze water samples, the aptasensor presented acceptable recovery rates of 93.1–100.1% and coefficients of variation (CVs) of 2.2–10.2%. In conclusion, the fluorescence aptasensor can accurately and sensitively detect SQX in water samples and has good application prospects.

## 1. Introduction

Sulfaquinoxaline (SQX) is a sulfonamide widely used to prevent infections, treat animals, and promote animal growth [1,2]. However, extra SQX can be released into the environment via animal feces and urine, causing significant water contamination [3]. The SQX in water can be directly drunk or enter the food chain through aquatic organisms and crops, eventually causing human allergies, affecting intestinal homeostasis, threatening thyroid and kidney health, and even triggering carcinogenicity [4,5,6]. For human health, various analytical methods are currently available for the detection of SQX, including liquid chromatography-tandem mass spectrometry [7], high-performance liquid chromatography (HLPC) [8], capillary zone electrophoresis [9], and enzyme-linked immunosorbent assay [10]. Although these methods are highly accurate, they have the disadvantages of requiring expensive equipment and complicated sample handling methods or being unstable. Thus, a reliable and low-cost analytical approach for detecting SQX in the environment is required.

Aptamers are single-stranded DNAs or RNAs that can fold into specific structures and stably bind to their targets via hydrogen bonds, van der Waals interactions, electrostatic forces, and so on [11,12]. Aptamers have some advantages over antibodies, such as low cost, easy modification, and high stability [13]. However, in aptamers, not every sequence region is involved in the binding reaction [14]. These unnecessary regions not involved in the binding can interfere with the reaction [15], and their removal can increase the affinity of the aptamer [16]. Attempts to shorten the sequences of aptamers based on their secondary structure have been previously proposed. For example, Shi et al. developed an SQX-specific aptamer after two truncations [4]. Although a new aptamer with a higher affinity was obtained, the method was cumbersome and required multiple cuts [4,17]. Therefore, an efficient and straightforward aptamers truncation approach is required. Retaining the conserved sequence of aptamers is considered an efficient truncation method [15]. Additionally, molecular docking and molecular dynamics (MD) simulations can be used to evaluate truncation results. Molecular docking can evaluate the interactions between ligands and receptors and predict their binding modes and affinities [18,19]. Meanwhile, MD simulations are used to evaluate the stability of the system [20]. Based on MD results, the binding energy can be decomposed and analyzed by the Molecular Mechanics-Poisson Boltzmann Surface Area (MM-PBSA) approach [21]. Herein, we used molecular docking and MD simulations to develop a novel method to truncate and characterize aptamers.

Gold nanoparticles (AuNPs) have been considered promising nanomaterials for biosensing [22], biological imaging [23], and catalysis [24]. Due to their high chemical stability, large specific surface area, and simple synthesis, AuNPs have been widely used in bioassays [25]. Particularly, the use of AuNPs in aptasensors to detect targets by colorimetry has been widely reported, but its sensitivity remains low [26]. Moreover, the absorption spectrum of AuNPs overlaps with the emission spectrum of Rhodamine B (RhoB). Therefore, AuNPs have the clear ability to quench the RhoB fluorescence [26], which can be used to develop highly sensitive fluorescent aptasensors. However, the use of a fluorescent aptamer based on AuNPs and RhoB for detecting SQX has not been previously reported.

In the present study, we obtained an aptamer against SQX by analyzing its conserved sequence and the results of molecular docking and MD simulations. The affinity of our aptamer was much higher than the aptamer developed by Shi et al. [4]. Then, we analyzed the interactions between SQX and the aptamer and demonstrated the high affinity and specificity of the aptamer. Finally, we developed a fluorescent aptasensor based on this aptamer and AuNPs to accurately detect SQX in water samples, which presented great potential for practical applications.

## 2. Materials and Methods

### 2.1. Reagents

SQX-specific aptamers have been previously screened by our research group [4]. SQX, sulfameter (SME), sulfamethazine (SMZ), sulfadimethoxine (SDM), sulfamonomethoxine (SMM), HAuCl_4_·4H_2_O, and graphene oxide were obtained from Sigma-Aldrich (St. Louis, MO, USA). Chlortetracycline (CTC), oxytetracycline (OTC), ofloxacin (OFL), and chloramphenicol (CAP) were obtained from Aladdin Biotechnology Inc. (Shanghai, China). RhoB and oligonucleotides were obtained from Sangon Biotechnology Inc. (Shanghai, China). Deionized water was purified using Millipore Milli-Q Ultrapure Water System (Bedford, MA, USA) and used for all experiments. DNA oligonucleotide stock solutions were prepared with binding buffer. All other chemicals and reagents were of analytical grade and obtained from Beijing Chemical Reagent Company (Beijing, China).

### 2.2. Molecular Docking

To analyze the homology of the aptamers, a phylogenetic tree was constructed using DNAMAN 8 (Lynnon Biosoft, San Ramon, CA, USA). The chemical structure of SQX was drawn by ChemDraw 19.0 (CambridgeSoft, Cambridge, MA, USA) and transformed into a three-dimensional structure by Chem3D 19.0. For the 3D structure of aptamers, their secondary structures were first predicted and converted to a 3D structure by RNA-composer (http://rnacomposer.cs.put.poznan.pl/, accessed on 8 July 2021). Then the target molecule and aptamers were docked using Autodock Tools 1.5.6 (National Biomedical Computation Resource). Results were analyzed using PYMOL 2.3.2 (Schrödinger) and Discovery Studio (Accelrys Inc., San Diego, CA, USA).

### 2.3. Molecular Dynamic (MD) Simulations

The accuracy of molecular docking results was further verified by MD simulations, and the interaction between the aptamers and SQX was explored. The topology and parameters of the ligand (SQX) were provided by SwissParam. The 60 ns production procedure was performed by GROMACS to generate trajectories for analysis. The Root Mean Square Deviation (RMSD) was used to measure the stability of the trajectory. When the RMSD curve has an upward sloping trend, it indicates that the conformation of the system might undergo significant movement; when the RMSD curve smoothly oscillates near a certain height, it indicates that the system has reached equilibrium [27]. A system with a low RMSD value indicates small structural deviations and high stability [20]. Finally, the binding energy components were calculated using g_mmpbsa software.

### 2.4. Fluorescence Polarization-Based Binding Affinity and Specificity Assays

To evaluate the affinity of candidate aptamers, 1 µg/mL SQX was mixed with different concentrations of aptamers (0, 25, 50, 100, 200, and 400 nM), then incubated in binding buffer (20 mM Tris-HCl, 100 mM NaCl, 5 mM KCl, 2 mM MgCl_2_, 1 mM CaCl_2_, pH 7.5) for 30 min at room temperature in the dark. Next, graphene oxide was added to the mixed solution and incubated for 10 min. The solution was centrifuged at 13,000 rpm for 10 min. Finally, the fluorescence of the supernatant was measured with exciting/emission at 492/518 nm on a Varioskan™ LUX Multimode Microplate Reader. All experiments were carried out three times. A negative control without SQX was also included to assess nonspecific binding. The dissociation constant (K_d_) value was obtained by the nonlinear fitting of Origin 9.0 software (Northampton, MA, USA) and calculated according to the nonlinear regression equation Y = B_max_ × X/(K_d_ + X) [28]; where X represents the concentration of aptamer, Y represents the relative fluorescence intensity, and B_max_ represents the most binding sites. The smaller the K_d_ value, the higher the affinity. On the other hand, 200 nM aptamers were incubated with 1 µg/mL of SQX or other antibiotics (SME, SMZ, SDM, SMM, CTC, OTC, OFL, and CAP) to determine their specificity. The fluorescence was measured after adding graphene oxide, and the difference between them and the negative control was calculated.

### 2.5. Synthesis and Characterization of AuNPs

The AuNPs were prepared according to Grabar et al. with some modifications [29]. First, 0.5 g trisodium citrate was dissolved in 50 mL dd∙H_2_O, and a 0.22 μm syringe filter was used to filter the solution to obtain a 1% trisodium citrate solution. Then, 1 mL 1% trisodium citrate was mixed with 50 mL 0.01% HAuCl_4_·4H_2_O at a boiling state. The mixture was heated until it became wine-red, then chilled to 25 °C. The final solution was stored at 4 °C in dim light. The absorption peak at 520 nm of the resultant AuNPs was measured using a VarioskanTM LUX Multimode Microplate Reader (Thermo Fisher Scientific Inc., Waltham, MA, USA).

### 2.6. Interactions between the Components of the Aptasensor

The prepared AuNPs (180 µL) were reacted with 60 nM SBA28-1, 20 mM NaCl and/or 1 µg/mL SQX in different combinations at room temperature. Then, the absorbances at 520 nm (A_520_) and 695 nm (A_695_) were determined for AuNPs, AuNPs-SQX, AuNPs-NaCl, AuNPs-SBA28-1, AuNPs-NaCl-SBA28-1, and AuNPs-NaCl-SBA28-1-SQX, and the A_695_/A_520_ ratio was calculated. The dynamic light scattering (DLS) of AuNPs in the above combinations was measured by a dynamic laser light scattering particle size analyzer. The fluorescence intensity of various combinations (RhoB, AuNPs-RhoB, SBA28-1-AuNPs-NaCl-RhoB, SQX-SBA28-1-AuNPs-NaCl-RhoB, SQX-RhoB, NaCl-RhoB, SBA28-1-RhoB, SBA28-1-SQX-RhoB, SBA28-1-NaCl-RhoB, SBA28-1-SBA28-1-NaCl-RhoB, AuNPs-RhoB-SQX, AuNPs-RhoB-SBA28-1, AuNPs-RhoB-SBA28-1-SQX) was determined under the excitation wavelength of 520 nm, and the ΔF was calculated using the formula ΔF = F − F_0_ (F is the fluorescence intensity in the presence of SQX, F_0_ is the fluorescence intensity in the absence of SQX).

### 2.7. Optimization of Detection Conditions

Next, we optimized the detection conditions to achieve high sensitivity with the established aptasensor. Briefly, 20 mM NaCl was added to 180 µL AuNPs and incubated for different times (0, 2, 4, 6, 8, and 10 min). The absorbance of the samples at 520 and 695 nm was detected using a Varioskan™ LUX Multimode Microplate Reader (Thermo Fisher Scientific Inc., Waltham, MA, USA). The highest A_695_/A_520_ ratio corresponded to the optimal NaCl time. Different doses of NaCl (0, 10, 20, 30, and 40 mM) were also evaluated. The ideal concentration was determined by the greatest A_695_/A_520_ ratio. Additionally, different concentrations of SBA28-1 (0, 20, 40, 60, 80, and 100 nM) were mixed with AuNPs and incubated for 30 min, and incubated with the optimal concentration of NaCl (determined above) for 6 min. Finally, SBA28-1 was incubated with or without SQX for 30 min, then AuNPs and NaCl were sequentially added to the solution for incubation, and different concentrations of RhoB (0, 20, 40, 60, 80, and 100 nM) were added to the above solution for fluorescence detection. The excitation and emission wavelengths were 520 nm and 577 nm, respectively. The fluorescence intensities of the SQX (F) and blank (F_0_) samples were measured, and the value of ΔF was calculated according to the formula; ΔF = F − F_0_.

### 2.8. Fluorescence Assay for SQX Detection

To verify the sensitivity of the aptasensor, different concentrations of SQX (0, 1.25, 2.5, 5, 10, 20, 40, 80, 160, 320, 640, 1000 ng/mL) were incubated with 60 nM SBA28-1 for 30 min, then 180 µL AuNPs were added for 30 min. After the reaction was completed, 20 mM NaCl was added to this mixture and reacted for 6 min, and 3 μM RhoB was added for 10 min. The fluorescence intensity (F, F_0_) was measured by the Varioskan™ LUX Multimode Microplate Reader (Thermo Fisher Scientific Inc., Waltham, MA, USA), and the ΔF (ΔF = F − F_0_) was calculated. To verify the specificity of the aptasensor, SQX-like antibiotics were tested, including SME, SMZ, SDM, SMM, CTC, OTC, OFL, and CAP. The concentration of total SQX-like antibiotics was 1 µg/mL, and the detection method was the same as above.

### 2.9. Analysis of Real Samples

Finally, the aptasensor was applied to detect lake and tap water samples from a farm to demonstrate its practicability and accuracy. First, the water sample was previously confirmed to be free of SQX compounds using HPLC. SQX was spiked in the samples to the final concentrations of 50, 100, and 150 ng/mL. To prepare the spiked water samples, the supernatants were filtered using a 0.22 μM syringe filter after centrifugation at 10,000 rpm for 10 min, then spiked with the different concentrations of SQX. Finally, the samples were analyzed with the above aptasensor. The recovery was calculated as follows: recovery rates = (concentration/spiked concentration) × 100% [28].

## 3. Results

### 3.1. Truncation Strategy

The aptamers used in the present study were previously screened by our research group [4]. According to the DNAMAN (Lynnon Biosoft, San Ramon, CA, USA) results, the sequences belong to six families, and their proportions were 28.9, 21.1, 10.5, 18.4, 15.8, and 5.3% (Appendix A). These families have common sequences that are beneficial for subsequent aptamer truncations. Family I has common sequences such as “GTC” (Appendix A). Other common sequences were detected in Family II, such as “TGC”. In family III, “TGA”, “TAA”, “GAG” were detected; in family IV, “ATG”, in family V, “AGG”; and, in family VI, “ACA”, “CTT” (Appendix A). These conserved motifs might bind SQX, and the stem-loop is an important structure for aptamers to recognize targets [13]. Hence, aptamers with conserved sequences on the stem-loop were retained as suitable for further investigation: SBA25 (Family I), SBA40 (Family II), SBA24 (Family III), SBA2 (Family IV), SBA28 (Family V), and SBA16 (Family VI) were identified as suitable aptamers for further investigation. On the premise of preserving the conserved structure of the aptamers as much as possible, they were truncated into short sequences: SBA25-1 (TGACCCTAAAGTCA), SBA40-1 (GGCTGACTGCT), SBA24-1 (GCGTGAGC), SBA2-1 (CAACCCCCATGTTG), SBA28-1 (CCCTAGGGG), and SBA16-1 (GCTGGTTTACCTTGC).

### 3.2. Molecular Docking between Aptamers and SQX

In this study, the binding mechanism between SQX and six candidate aptamers was evaluated using Autodock Tools 1.5.6. The binding energy between SQX and SBA25-1, SBA2-1, and SBA16-1 were −2.60, −1.55, and −2.84 kcal/mol, respectively, significantly smaller than the binding energy between SQX and SBA40-1, SBA24-1, and SBA28-1 (−3.76, −3.56, and −4.01 kcal/mol, respectively). Hence, SBA25-1, SBA2-1, and SBA16-1 were not used for subsequent experiments. On the other hand, the binding energies of SBA40-1, SBA24-1, and SBA28-1 were stronger than the original sequence (the binding energy between SQX and SBA40, SBA24, and SBA28 were −1.42, −1.21, and −1.33 kcal/mol, respectively). Therefore, the truncation method proposed was effective. The molecular docking is shown in Figure 1. The molecular docking results showed that the binding sites between SQX and SBA40-1 were mainly related to the bases C-7, T-8, G-9, and C-10, and SQX was located inside its loop (Table 1). Four hydrogen bonds were formed between SQX and C-7, T-8, and G-9 of SBA40-1, and one π-sulfur and one π-π T-shaped interaction were formed between SQX and G-9 of SBA40-1. Additionally, a π-Anion was formed between SQX and C-10 of SBA40-1. Moreover, SQX was wrapped in the loop of aptamer SBA24-1 by four deoxyribonucleotides (G-1, C-2, G-3, and G-5), two hydrogen bonds (between SQX and G-3 and G-5), and two π-π T-shaped interaction (between SQX and G-1 and G-2). SQX was also wrapped in the loop of SBA28-1 by three deoxyribonucleotides (A-5, G-6, and G-7), four hydrogen bonds (between SQX and A-5, G-6, and G-7), three π-sulfur (between SQX and A-5 and G-6) and two π-π T-shaped interactions (between SQX and G-6).

### 3.3. MD Simulations of Aptamer-Target Interactions

Next, we performed MD simulations to verify the SQX and aptamer molecular docking results. After a 60 ns production simulation of the SQX/aptamers complex, the deviation of the structure from the initial configuration was monitored by the RMSD to measure the system’s stability. The RMSD curve of the SQX/SBA28-1 and SQX/SBA40-1 complexes rose slowly from 0 to 15 ns (Figure 2a,c), which suggested that the skeleton of the crucial nucleic acid was moving at this time. During the 15 ns–60 ns interval, the SQX/SBA28-1 complex was balanced since the RMSD curve only fluctuated around 6 Å, and the amplitude stayed within 2 Å. The SQX/SBA40-1 complex was also balanced (the RMSD curve only fluctuated around 4.5 Å, and the amplitude stayed within 2 Å). However, the SQX/SBA24-1 complex was not balanced until 35 ns (the RMSD curve fluctuated around 7.5 Å) (Figure 2b). These results indicated that SBA28-1 and SBA40-1 have higher stability in binding to SQX than to SBA24-1.

The K_d_ values of SBA40-1, SBA24-1, and SBA28-1 measured using graphene oxide were 36.73, 74.19, and 27.36 nM (Figure 3a), respectively, lower than the original aptamers (198.26, 272.25, and 170.52 nM, respectively) and the aptamer developed by Shi et al. (82.54 nM) [4]. These results demonstrated the effectiveness of molecular docking and MD simulations for aptamer truncation. The specificity of the aptamers was also determined using graphene oxide (Figure 3b). Considering the affinity and specificity of aptamers, SBA28-1 was used for the following experiments. The binding energy of the SQX/SBA28-1 complex was decomposed using g-mmpbsa. Based on the Molecular Mechanics/Poisson Boltzmann (Generalized Born) Surface Area method, the binding energy of the SQX/SBA28-1 complex (−17.995 kJ/mol) was composed of Van der Waals (−20.594 kJ/mol), electrostatic (−3.432 kJ/mol), polar solvation (8.209 kJ/mol), and nonpolar solvation (−2.176 kJ/mol) energies. The Van der Waals energy presented the major contribution to the total binding free energy. The polar solvation energy negatively affected the total binding free energy, in contrast to the Van der Waals, electrostatic, and nonpolar solvation energies.

### 3.4. Principle of the Aptasensor

Next, we developed an aptasensor using the SQX28-1 aptamer to detect SQX quantitatively. The sensing mechanism is illustrated in Figure 4. Without SQX, the AuNPs are expected to adsorb SBA28-1 and form AuNPs-SBA28-1 complexes, which allows the good dispersion of AuNPs in high concentrations of NaCl. In this case, the high fluorescence of RhoB is effectively quenched by the dispersed AuNPs [30,31]. In the presence of SQX, the aptamer can attach to SQX rather than the AuNPs, and this well-folded binding complex is more stable than the AuNPs-SBA28-1. Thus, the AuNPs form aggregates in the presence of NaCl and fail to quench RhoB fluorescence [32,33]. Therefore, the content of SQX in the sample can be detected by the fluorescence of RhoB.

### 3.5. Interactions between the Components of the Aptasensor

Further, we evaluated the absorption spectral responses of AuNPs under different conditions to confirm the feasibility of the proposed aptasensor (Figure 5a). The AuNPs have an absorption peak at 520 nm, and the addition of SQX or SBA28-1 did not change this absorption. Since NaCl can significantly aggregate AuNPs, the absorption peak of AuNPs consequently shifts from 520 to 695 nm with the addition of NaCl. Moreover, the AuNPs can bind to SBA28-1 and stabilize without SQX, implying that AuNPs would disperse again in the presence of NaCl, and the absorption spectrum would still present an absorption peak at 520 nm. However, in the presence of SQX, the aptamer reacted with SQX instead of binding to the AuNPs, so the AuNPs aggregated when NaCl was added. The aggregated AuNPs form a new UV absorption peak at 695 nm.

The diameter of AuNPs was approximately 0.60 nm (Appendix A), and the addition of SQX did not significantly affect the diameter (Appendix A). When SBA28-1 was introduced and adsorbed onto the surface of AuNPs without NaCl, the diameter of AuNPs grew to 1.70 nm (Appendix A). In the presence of NaCl, the diameter of AuNPs increased to 64.72 nm (Appendix A). However, in the presence of SBA28-1, the AuNPs were no longer aggregated by NaCl, and their diameter was reduced (Appendix A). After SQX was added to the solution, the AuNPs aggregated again and the diameter enlarged (Appendix A).

Then, the fluorescence intensity was determined to confirm the sensing feasibility. RhoB alone had the highest fluorescence intensity, which was lowered by adding AuNPs (Figure 5b). After adding NaCl and SBA28-1 to the solution, the fluorescence intensity of RhoB was almost constant and improved again when SQX was added. The fluorescence intensity of RhoB was not affected by SQX, NaCl, and SBA28-1, nor by the binging of SQX and SBA28-1 (Appendix A). Similarly, when AuNPs were in the system, the binding of SQX and SBA28-1 did not affect the fluorescence intensity of RhoB without NaCl (Appendix A).

### 3.6. Optimization of Detection Conditions

To optimize the aptasensor, the incubation time with NaCl and the concentrations of NaCl, SBA28-1, and RhoB was evaluated. First, the influence of incubation time with NaCl on detection performance was investigated (Appendix A). As the incubation time with NaCl increased, the A_695_/A_520_ ratio enhanced until 6 min, then stabilized. Thus, the final incubation time with NaCl was determined as 6 min. The A_695_/A_520_ ratio was maximum at 20 mM NaCl (Appendix A), which was considered the optimal NaCl concentration. The A_695_/A_520_ ratio was continuously decreased until the concentration of SBA28-1 was 60 nM (Appendix A). Thus, 60 nM SBA28-1 was selected for subsequent experiments. Finally, the ΔF continued to increase until the concentration of RhoB reached 3 µM, considered the optimal RhoB concentration (Appendix A).

### 3.7. Properties of the Aptasensor

Furthermore, the ΔF increased in the linear range of 1.25–160 ng/mL (Figure 6a). The linear regression equation was ΔF = 0.1022 C_SQX_ + 1.1479 (R^2^ = 0.9942). The limit of detection (LOD) for SQX was 1.04 ng/mL (LOD = 3 × SD/S; where S represents the slope of the calibration curve, and SD is the standard deviation for the blank). Therefore, the aptasensor presented excellent sensitivity for SQX.

Next, eight structurally related or unrelated antibiotics (SME, SMZ, SDM, SMM, CTC, OTC, OFL, and CAP) were used to study the specificity of the aptasensor for SQX. The ΔF of the aptasensor for other antibiotics, such as SME, was significantly smaller than that of SQX (Figure 6b), showing that the aptasensor has good specificity against SQX.

### 3.8. Validation of the Aptasensor

Based on the optimal reaction conditions determined above, different concentrations of SQX (50, 100, and 150 ng/mL) were added to SQX-free lake water and tap water to validate the aptasensor. Each concentration of SQX was analyzed in five replicates (*n* = 5). The detection data showed a good recovery rate in the range of 96.1 to 100.1%, and the CVs ranged from 2.2 to 10.2% (Table 2). Altogether, these results demonstrated the accuracy of this detection platform for SQX in water samples.

## 4. Conclusions

In summary, we successfully truncated the aptamers and selected SBA28-1 (CCCTAGGGG) due to its higher affinity (K_d_ = 27.36 nM). The truncated approach was validated by molecular docking and MD simulations. The effects of Van der Waals, electrostatic, and solvation energies on target recognition by the SBA28-1 aptamer were also explored. We found that the Van der Waals energy had a major contribution to the recognition of SQX by SBA28-1. Finally, based on the FRET between AuNPs and RhoB, we established a sensitive and accurate platform to detect SQX in water samples. The linear range of detection for the aptasensor was 1.25–160 ng/mL, and the detection limit was 1.04 ng/mL. The recovery range was 93.1–100.1%, and the CVs were 2.2–10.2%. Therefore, the aptasensor has excellent application prospects in water testing.

## Figures and Tables

**Figure 1 biosensors-12-00513-f001:**
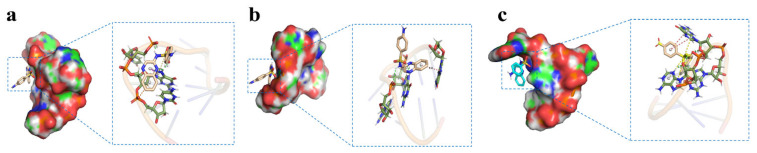
Predicted binding between SQX and aptamer as indicated by the surface display diagram and molecular docking (**a**) SBA40-1; (**b**) SBA24-1; (**c**) SBA28-1.

**Figure 2 biosensors-12-00513-f002:**
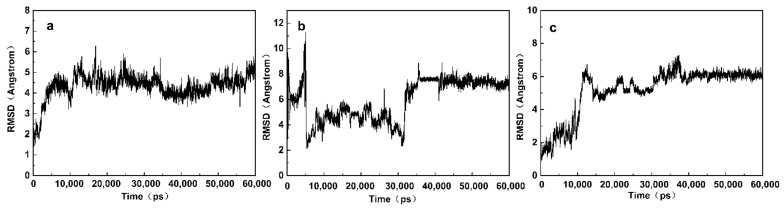
RMSD curve of (**a**) the SBA40-1 aptamer/SQX complex, (**b**) the SBA24-1 aptamer/SQX complex, and (**c**) the SBA28-1 aptamer/SQX complex.

**Figure 3 biosensors-12-00513-f003:**
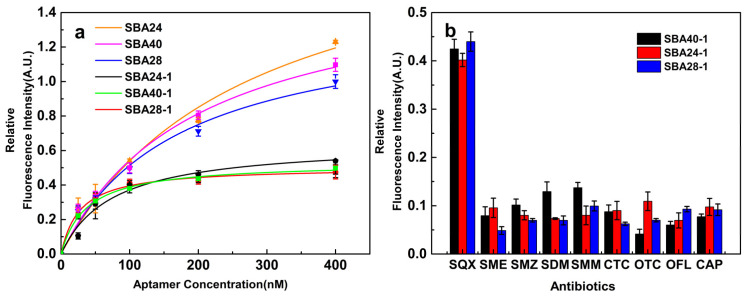
(**a**) Correlation between FAM-aptamer concentration and relative fluorescence intensity; (**b**) Relative fluorescence intensity of different structural antibiotics relative to SQX measured as the reference standard.

**Figure 4 biosensors-12-00513-f004:**
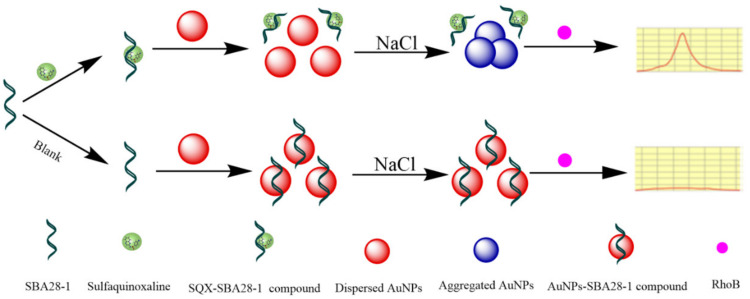
Schematic illustration of the fluorescent aptasensor for detecting SQX based on label-free aptamer and the FRET between RhoB and AuNPs.

**Figure 5 biosensors-12-00513-f005:**
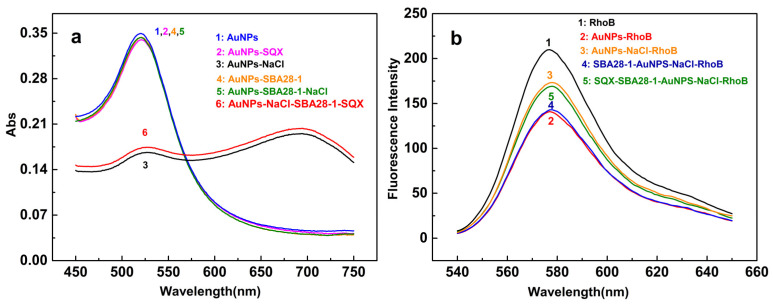
(**a**) Absorption spectra of AuNPs solutions in different samples; (**b**) Fluorescence spectra of RhoB in different samples.

**Figure 6 biosensors-12-00513-f006:**
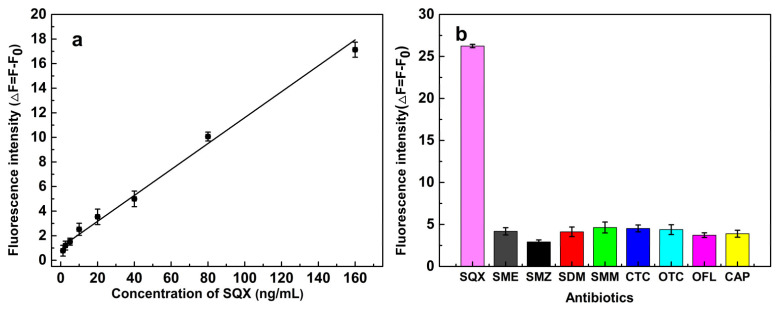
(**a**) Linear fitting of the fluorescence intensity of the aptasensor; (**b**) Selectivity of the fluorescent aptasensor for SQX and other antibiotics: SME, SMZ, SDM, SMM, TET, OTC, OFL, and CAP.

**Table 1 biosensors-12-00513-t001:** Intermolecular interactions between DNA and the small molecules.

Aptamer	Binding Energy (kcal/mol)	The Base that Binds to the Ligand	The Lengths of the Hydrogen Bond (Å)	The Lengths of the π-Sulfur/π-π T-Shaped/π-Anion (Å)
SBA 40-1	−3.76	C-7	2.90	-
		T-8	1.99	-
		G-9	2.05/2.45	4.51/5.10
		C-10	-	4.59
SBA 24-1	−3.56	G-1	-	4.79
		C-2	-	4.83
		G-3	2.30	-
		G-5	2.84	-
SBA 28-1	−4.01	A-5	1.81/2.05	5.92/5.32
		G-6	1.79	4.16/4.88/4.84
		G-7	2.68	-

“-”: no bond.

**Table 2 biosensors-12-00513-t002:** Mean Recoveries and Coefficients of Variation for the SQX in Water Using Optimized Fluorescence (*n* = 5).

Sample	Spiked Concentration (ng/mL)	Mean Recovery (%) ± SD	CV(%)
Lake Water	50	96.1 ± 9.8	10.2
	100	96.2 ± 6.2	6.4
	150	98.4 ± 2.1	2.2
Tap Water	50	93.1 ± 4.5	4.9
	100	98.8 ± 6.1	6.2
	150	100.1 ± 3.3	3.3

SD: standard deviation; CV: coefficient of variation.

## Data Availability

Not applicable.

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
