# Peer review of "An AuNPs-Based Fluorescent Sensor with Truncated Aptamer for Detection of Sulfaquinoxaline in Water"

_biosensors, 2022, doi:10.3390/bios12070513_

Round 1
Reviewer 1 Report
General comment: The study “A label-free turn-on fluorescent sensor based on truncated ap-2 tamer for detection of sulfaquinoxaline” from Chen et al. explores an interesting topic. But in order to increase the usefulness and significance of the study, it needs a revision before being considered suitable for readers. I think that the paper should be improved in these aspects:
1. The title of the manuscript does not give the proper meaning of the research work, authors should update it.
2. In the abstract section authors maintained the detection range (line 23) of 1.25-160 ng/mL,…. How do authors justify the accuracy of these values, please explain.
3. In page 1 line 24, 93.1-100.1% recovery rate…. It is hard to accept these values, please check it again.
4. In the introduction section, address the aim of your investigation, before the final paragraph.
5. Certain things in the manuscript have not been seeming properly. The authors are suggested to check the entire manuscript and do the required changes as per the journal format.
6. In page 2, line 83, HauCL4.4H2O, please update in a proper format.
7. In the conclusion section add more key findings of your investigations.
8. Overall, the manuscript seems good, but some minor grammatical and typological errors can be seen, please the entire manuscript by any native English speaker.
Reviewer 2 Report
In the reviewed manuscript, the authors described an interesting technique for truncating aptamers with a unique stem-loop structure and determined the binding mechanism and energy composition between aptamers and sulfaquinoxaline (SQX). It is amazing and not very realistic to record the percentage of the aptasensor's profit in water samples greater than 100% - such notation (100.1%) must be corrected. The English in the present manuscript is not of publication quality and require major improvement.
Reviewer 3 Report
In this manuscript, Chen et al. developed an aptamer-based fluorescence sensor for sulfaquinoxaline (SQX) detection by controlling rhodamine B (RhoB) signal turn-on with the AuNP/NaCl aggregation induced change of absorption wavelength. The authors carefully characterized stepwise truncation of the SQX aptamer candidates using molecular docking predication, molecular dynamics simulation, and graphene oxide binding assays. Then, the authors studied the spectral property, detection range, target selectivity of this sensing system and showed satisfying performance. The authors further applied the sensor for spiked SQX detection in water samples and showed the potential of applications.
The experimental design in this work is straightforward, and the data are carefully documented. However, few interpretations of the results should be reconsidered to ensure the quality of this work. This paper should be re-evaluated for acceptance in Biosensors after addressing following concerns.
Line 61, the abbreviation MM-PBSA should be specified.
Line 200-201, "... identified as suitable aptamers for further investigation, considering their homology, enrichment, and secondary structure." The criteria of homology, enrichment, and secondary structure when choosing candidates is not clear, which should be specified in more details. (e.g., length of homologous nucleotides, percentage of enrichment, types and number of structures, free energy, etc.)
What are the excitation and emission wavelengths used for the fluorescence assays in Figure 5 and Figure 6?
Line 277, when AuNPs are synthesized using citrate as the capping agent, the overall surface charge could be negative. However, the AuNP itself should be positively charged. As the authors suggested in line 272-273, the AuNPs will absorb SBA28-1 aptamer to form complexes. The aptamer DNAs are negatively charged. If the AuNPs are also negatively charged, this absorption process will be ineffective, and the sensor design will not work. The authors should provide Zeta-potential data as a support if they claim their AuNPs (bare particle without aptamer binding) are negatively charged.
Line 274-279, the authors suggest that the fluorescence turn-on of the sensor is caused by less effective quenching of RhoB emission when SQX is introduced and bound to the aptamer on AuNP surface. However, the maximum emission of RhoB is close to 580 nm as shown in Figure 5B and Figure S9. The absorption of the AuNPs-NaCl-SBA28-1-SQX system (group 6 in Figure 5A) at 580 nm is increased compared with the group 5 without SQX target, meaning a more effective quenching of RhoB emission upon SQX addition. This result is against the proposed model by the authors. A better interpretation of the data will be that the decrease of AuNPs' absorption at 520 nm region upon SQX addition (group 6 vs group 5 in Figure 5A) led to less absorption of the excitation light during the fluorescence assays and resulted in more effective RhoB excitation and higher emission (signal turn-on).
Line 289, from Figure 5A, the absorption peak of AuNPs shifts from 520 to around 660 nm, rather than 695 nm. Why is 695 nm specifically chosen for later assays?
Figure 5B, the AuNPs-RhoB-NaCl group should also be shown.
Round 2
Reviewer 1 Report
Accept
Reviewer 3 Report
All the concerns are properly addressed. This manuscript is in good quality for publication.